# Social and Contextual Influences on Antibiotic Prescribing and Antimicrobial Stewardship: A Qualitative Study with Clinical Commissioning Group and General Practice Professionals

**DOI:** 10.3390/antibiotics9120859

**Published:** 2020-12-01

**Authors:** Aleksandra J. Borek, Sibyl Anthierens, Rosalie Allison, Cliodna A. M. Mcnulty, Philip E. Anyanwu, Ceire Costelloe, Ann Sarah Walker, Christopher C. Butler, Sarah Tonkin-Crine

**Affiliations:** 1Nuffield Department of Primary Care Health Sciences, University of Oxford, Oxford OX2 6GG, UK; christopher.butler@phc.ox.ac.uk (C.C.B.); sarah.tonkin-crine@phc.ox.ac.uk (S.T.-C.); 2Department of Family Medicine and Population Health, University of Antwerp, 2610 Antwerp, Belgium; sibyl.anthierens@uantwerpen.be; 3Primary Care and Interventions Unit, Public Health England, Gloucester GL1 1DQ, UK; rosie.allison@phe.gov.uk (R.A.); cliodna.mcnulty@phe.gov.uk (C.A.M.M.); 4Department of Primary Care and Public Health, Imperial College London, London W6 8RP, UK; anyanwup@cardiff.ac.uk (P.E.A.); ceire.costelloe@imperial.ac.uk (C.C.); 5School of Medicine, College of Biomedical and Life Sciences, Cardiff University, Cardiff CF14 4XN, UK; 6National Institute for Health Protection Research Unit in Healthcare Associated Infections and Antimicrobial Resistance, Oxford OX3 9DU, UK; sarah.walker@ndm.ox.ac.uk; 7National Institute for Health Research Biomedical Research Centre, Oxford OX3 9DU, UK; 8Nuffield Department of Medicine, University of Oxford, Oxford OX3 9DU, UK

**Keywords:** antibiotic prescribing, antimicrobial stewardship, primary care, qualitative

## Abstract

Antibiotic prescribing in England varies considerably between Clinical Commissioning Groups (CCGs) and general practices. We aimed to assess social and contextual factors affecting antibiotic prescribing and engagement with antimicrobial stewardship (AMS) initiatives. Semi-structured telephone interviews were conducted with 22 CCG professionals and 19 general practice professionals. Interviews were audio-recorded, transcribed, and analyzed thematically. Social/contextual influences were grouped into the following four categories: (1) Immediate context, i.e., patients’ social characteristics (e.g., deprivation and culture), clinical factors, and practice and clinician characteristics (e.g., “struggling” with staff shortage/turnover) were linked to higher prescribing. (2) Wider context, i.e., pressures on the healthcare system, limited resources, and competing priorities were seen to reduce engagement with AMS. (3) Collaborative and whole system approaches, i.e., communication, multidisciplinary networks, leadership, and teamwork facilitated prioritizing AMS, learning, and consistency. (4) Relativity of appropriate prescribing, i.e., “high” or “appropriate” prescribing was perceived as relative, depending on comparators, and disregarding different contexts, but social norms around antibiotic use among professionals and patients seemed to be changing. Further optimization of antibiotic prescribing would benefit from addressing social/contextual factors and addressing wider health inequalities, not only targeting individual clinicians. Tailoring and adapting to local contexts and constraints, ensuring adequate time and resources for AMS, and collaborative, whole system approaches to promote consistency may help promote AMS.

## 1. Introduction

Overuse of antibiotics is one of the main contributors to antimicrobial resistance. Because most antibiotics in England are used in the community [1], many with no benefit to patients, more prudent prescribing in primary care is needed. Many different antimicrobial stewardship (AMS) initiatives have been used to optimize antibiotic prescribing in primary care [2,3]. AMS strategies are frequently targeted directly at and provided to individual prescribers (e.g., diagnostic scores and training) and some have been implemented nationally (e.g., prescribing targets). For example, the Quality Premium (QP) is an incentive scheme to improve the quality of primary care services in England and has included targets for Clinical Commissioning Groups (CCGs) to optimize antibiotic prescribing [4]. The CCGs are organizations responsible for commissioning and ensuring quality of primary care services in the English National Health Service (including funding and monitoring the quality of services offered in general practices). The introduction of the QP antibiotic targets was associated with reductions in antibiotic prescribing [5,6], and over 83% of CCGs (between 2015 and 2018) met the QP targets for reducing total and broad-spectrum antibiotic prescribing in primary care [4]. However, not all CCGs and general practices reduced antibiotic prescribing at the same rate or met the QP targets. There was also considerable overall variation in the volume of antibiotic prescribing [7,8,9,10,11,12], i.e., two-fold differences between the lowest- and highest-prescribing CCGs (2010–2017) [7] and general practices (2004–2005) [12]. 

Some variation in antibiotic prescribing can be explained by patient-related factors, such as comorbidities, smoking, age, weight, ethnicity, or socioeconomic status [7,8,10,12]. However, substantial variation between practices remains even after accounting for comorbidities, smoking, and deprivation [9,10]. Higher antibiotic prescribing has been associated with practice characteristics, such as location (north of England, rural), deprivation, larger practice size, or high consultation rates for respiratory infections [7,8,9,11,12,13]. Differences in antibiotic prescribing may also be associated with prescribers’ characteristics (e.g., higher prescribing by male, older GPs and those qualified outside the UK) [12]. 

Antibiotic prescribing is undoubtedly influenced by individual-level factors, both patient-related (e.g., clinical presentation, history, and comorbidities) and clinician-related (e.g., knowledge of guidelines, experience, and risk tolerance). However, these are not the only influences. There have been increasing calls to recognize and address social and contextual influences on antibiotic prescribing and AMS [14,15]. Antibiotic prescribing is a process influenced by the interaction between individuals and the social and physical environment [15], and antimicrobial resistance is “deeply social, shaped by cultural, political, and economic processes” so it requires a social solution (p.1) [14]. A review of international, qualitative studies on general practitioners’ (GPs) experiences of antibiotic prescribing for acute respiratory tract infections (RTIs) concluded that clinicians are influenced by the intrapersonal, interpersonal, and contextual situation [16]. The examples of contextual influences included considering patient retention and financial factors, medico-legal issues (i.e., concerns about being sued for “missing something” in a patient), continuity of care, work pressure and fatigue, over-the-counter access to antibiotics, lack of consistent national guidelines, and incentives from the pharmaceutical industry [16]. When comparing views of prescribers from low- and high-prescribing practices in England, some of the differences in contextual influences on antibiotic prescribing included perceptions of others’ prescribing and support for no-antibiotic decisions, consultation length, triage system, access to microbiologists or pharmacists, and (for all practices) consequences of complaints or adverse events [17]. While several qualitative studies have described influences on antibiotic prescribing, some of which are contextual, they have not specifically explored how social and contextual factors influence engagement with AMS interventions.

We aimed to describe which social and contextual influences were perceived by CCG professionals and general practice professionals as impacting on antibiotic prescribing and engagement with AMS, and how they impacted on antibiotic prescribing. This qualitative study was conducted with the overall aim to understand the implementation and mechanisms of the QP and related AMS interventions (reported previously [18]), as part of a larger study on improving implementation of AMS interventions (“STEP-UP” [19]). 

## 2. Methods 

### 2.1. Study Participants

CCG professionals responsible for AMS were identified from a list of contacts for all CCG Medicines Management/Optimisation teams. The initial recruitment used purposeful sampling to ensure diverse CCG characteristics (antibiotic prescribing rates, location, deprivation, and number of GP practices). However, due to a low response rate, eventually all 145 contacts were emailed study invitations. From the participating CCGs, 12 diverse CCGs were selected; within each CCG, practices with the highest and lowest antibiotic prescribing rates (top/bottom 25%) were selected, and 162 practices were invited. Invitations were sent by email or post, and non-responders were followed up twice. Participants were asked to return signed consent forms or give verbal consent which was recorded. The study was approved by the University of Oxford’s research ethics committee (ref. R53960) and the Health Research Authority (ref. 230479). 

### 2.2. Data Collection

Telephone interviews were conducted by the first author (A.B., a sociologist and experienced qualitative researcher) between December 2017 and September 2018 using two semi-structured topic guides; one for CCG and one for practice professionals (Appendix A). In brief, participants were asked about their role in addressing AMS, views on AMS and the QP, how they decided to address antibiotic prescribing targets, how they communicate within and between CCGs and general practices, and suggestions related to the QP and AMS. Open questions were followed with prompts to gather further detail. Participants were offered 40 GBD shopping vouchers or payment to their practice for their participation. The interviews were audio-recorded, transcribed verbatim, and transcripts were then checked for accuracy and anonymized.

### 2.3. Data Analysis

Interview transcripts were uploaded to NVivo software (v.11) and analyzed thematically [20]. One researcher (A.B.) coded all transcripts inductively line-by-line. Three researchers (S.A., R.A., S.T.C.) double-coded 20 transcripts (6–7 transcripts each), making notes on the key topics and potential themes. The analysis (i.e., coding, developing categories, and themes) was discussed with the team, and the codes and categories were refined and defined. The key themes (i.e., those that addressed the research question and were supported by the data) were identified, discussed, and agreed on with the study team. After analyzing the CCG and practice interviews separately, the findings were compared and combined into one integrated framework, with attention to similarities and disparities.

## 3. Results

Twenty-two CCG professionals and 19 general practice professionals were interviewed (Table 1). The CCG professionals worked across 33 CCGs, with four working across multiple CCGs. The CCGs and practices were diverse in terms of antibiotic prescribing rates, deprivation, and number of practices and prescribers. The interviews with CCG professionals lasted 35 to 65 (mean 52) minutes and with practice professionals 26 to 57 (mean 40) minutes. The overarching theme was social and contextual influences. Below, we describe the four categories of these influences (summarized in Figure 1) and how they were perceived to influence antibiotic prescribing and AMS. Additional quotes illustrating the findings are available in Appendix A. 

### 3.1. Immediate Context: Patient, Clinician, and Practice Characteristics

CCG professionals reported a considerable variation in antibiotic prescribing between practices, which reflected geodemographic and socioeconomic differences influencing patient health and practice characteristics. These complex and often historically rooted influences were seen as a challenge for reducing antibiotic prescribing. 

CCG and practice professionals described that patients in some areas had greater needs or expectations for antibiotics. In particular, higher deprivation was perceived as contributing to higher antibiotic prescribing by being linked to the following: more illness and comorbidities; less (ability to) self-care and less access to over-the-counter remedies; lower threshold for GP appointments and antibiotic treatment (i.e., patients consulting earlier in the illness, expecting antibiotics); and need for illness validation (e.g., sick notes) and quicker recovery. 


*The practice is in a deprived area and the patient expectation here is that they will get a prescription for an antibiotic. Not only that, but people live in overcrowded situations and there is cross-infection with wider infections etc. (…) I think there may be a clinical reason but not sufficient to explain the complete high prevalence but I feel we often get into a situation where the patient’s expectation is for a prescription for antibiotic and when we refuse or try and explain that it’s not appropriate, we get a very irate patient, and I can understand that because there is a pressure on the patient to get well and get back to work as their income levels are lower in the patch we work in. Also, children with coughs and colds, parents would expect antibiotics because their child staying at home would mean that the parent would have to take time off work and again, loss of income. And so a patient more or less expects a prescription. *
*(GP-7, high-prescribing)*

Patient expectations for antibiotics were generally seen as higher in more deprived areas, but a couple of CCG participants described expectations to be higher in less deprived, smaller, rural communities (“a reasonably affluent population who are quite demanding” (CCG-2)). Moreover, patients from different cultural/ethnic backgrounds were described as having higher expectations for prescriptions influenced by their health-related cultural norms, and being difficult to reassure without medication, especially when there were also language barriers. 


*…we have a large Pakistani community and I think culturally there is this expectation that when you go to the doctors, they’ll give you a prescription and sometimes it doesn’t even matter what that prescription is, but you have to have some medication and also there’s a language barrier quite often as well… and some of our patients aren’t particularly well educated as well, so trying to give people some understanding of even basic statistics or basic kind of biology is quite difficult and trying to explain to somebody that you don’t need a prescription.*
*(GP-10, low/medium-prescribing)*

Practice and staff characteristics were also perceived as influencing antibiotic prescribing and AMS. High-prescribing practices were described as those that “struggled as general practices” (CCG-5), those most under-pressure and “overstretched” (CCG-9) due to staff shortages and high demand (“utterly deluged with needing to see patients” (CCG-2)), undergoing transitions, and consequently with more changing, temporary staff. In such practices, there seemed to be less “ownership” of prescribing, less engagement with AMS, and higher antibiotic prescribing was seen as just one of many challenges (these practices “don’t perform well in other areas as well, so it’s not just antibiotics” (CCG-6)). Stable staff and patients, allowing for continuity of care, were seen as factors enabling more appropriate prescribing.


*We have good response from the practices that have stable staff. Unfortunately, there are quite a lot of practices that have a lot of locums, transient staff, maybe inexperienced staff and what we find is that in those areas and often areas that have high levels of chronic disease we tend to get a high prescribing rate and it’s difficult to address that particularly when you can’t engage the staff as they’re not there long-term. (…) There’s been a lot of retirement from practices in [area] and there’s been takeovers of practices and as a consequence you’ve had lots of locum staff or short-term staff there. I think once there’s less ownership of the problem, you tend to see a deterioration in the antibiotic prescribing rates. I would argue we could map that across the city and say, we know these practices have a problem and will continue because they are struggling as general practices and as a consequence that impacts on their antimicrobial prescribing.*
*(CCG-5, high-prescribing)*

Participants also reported that certain types of prescribers might be more likely to overprescribe antibiotics. They linked higher prescribing to (practices with more) locums, older GPs, non-GP or inexperienced prescribers, or those qualified outside the UK. For example, some older GPs (i.e., as compared with more recently trained/qualified) were seen as having a tendency (or habit) to prescribe more antibiotics, whereas some locums were seen as less responsible and accountable (with less “ownership”) for their prescribing (“just want to get through their patient list” (CCG-2)), less aware of local guidelines, and lacking helpful long-term relationships with patients.


*This is a bit controversial, but there’s quite a lot of locum GPs and there’s quite a lot of older GPs… some of the older GPs aren’t quite as up to date on the guidelines on what to prescribe so they tend to prescribe drugs that they’ve prescribed a lot in the past and the guys who come as locums don’t have that same kind of… it’s my permanent base so I’m very strict on what I prescribe and I stick to the guidelines, but some of the locums don’t and because they work across different places… and the guidelines may be slightly different.*
*(GP-11, high-prescribing)*

Individual-level factors were also perceived as sometimes increasing prescribing and disengagement with AMS (e.g., unawareness of inappropriate prescribing and lacking the motivation to change). Some higher prescribers were seen as more concerned about risks or having experience of adverse effects, and potential malpractice accusations and legal consequences, of not prescribing antibiotics. Therefore, some were described as prescribing defensively (i.e., in a risk-averse way) or feeling defensive about their prescribing which precluded engagement with AMS. This reflected general perceptions of insufficient importance of prudent antibiotic prescribing and more risk of negative consequences (to the patient and the prescriber) of not prescribing than of prescribing antibiotics. 


*People’s thresholds as to what’s reasonable can be subjective so it depends on how open-minded you are when you go into an audit. If you do an audit feeling defensive, then it’s probably a waste of time because you’re going to find an excuse to justify things (…) practices that are practicing defensively or under a lot of pressure seem to be the ones that are much less likely to change.*
*(CCG-9, multiple CCGs)*


*I still don’t think there is the level of gravity placed on inappropriate prescribing. It’s still perceived widespread as a sort of poor judgement call. If you imagine that we were prescribing drugs for diabetes or hypertension or other long-term conditions inappropriately, it would be seen as a fairly serious deficit in care. Whereas if it’s an antibiotic, it’s seen as poor judgement or just GPs taking the easy life.*
*(CCG-21, high-prescribing)*

### 3.2. Wider Context: Pressures on the System and Resources

The wider context of the healthcare system, with many competing priorities and inadequate resources, were perceived as hindering engagement with AMS. 

For most CCG professionals, AMS constituted a small part of their roles, which limited how they could support practices. Most considered the available resources (time, staff, and finance) insufficient for more time- and resource-intensive AMS work (e.g., auditing prescribing in practices, delivering face-to-face training, and offering financial incentives), although these were seen as more impactful than less intensive strategies (e.g., emailing prescribing targets and reports and providing online data). 

Similarly, practice professionals described AMS as an area of special interest for some rather than of concern for all practice professionals. They reported large workload and demand, staff shortages, and time pressures (including short consultations) as impeding their engagement with AMS and prudent antibiotic prescribing. In this pressured context, seeing patients was prioritized over engaging with AMS strategies (seen as requiring additional time), and prescribing antibiotics was seen as a way to deal with workload/demand (i.e., as easier and quicker than not prescribing and as limiting patients returning for antibiotics) and to help reduce workload elsewhere (e.g., in out-of-hours or emergency services). These challenges were seen as part of a larger problem of pressures on the NHS system. There was a sense that professionals are increasingly “tired” of the constant pressure to change practice but without the necessary changes in the context that affects it.


*The thing that everyone recognizes is that it takes so much longer to explain why you’re not prescribing than it does to just churn out a script for an antibiotic so time is key and over-pressurized practices with insufficient staff for the workload might be predicted to prescribe more or more often inappropriately, possibly. I think we need to keep an eye on the staffing and workload levels because having time with the patient is the most effective way of exploring their actual issues and what really is worrying them….*
*(GP-18, medium/low-prescribing)*


*So the issue really isn’t just simply about antimicrobials. It’s about antimicrobial prescribing within a constrained under-pressure system that in some areas is really very very delicate. Banging on and saying messages about the antibiotic problems ahead is very hard when you’ve got everyday pressures in general practices you’ve got to contend with and a waiting room full of people that you’ve got to get through. […] I think people are tired with change and tired of the relentless messages, do this, do that. So I think at the moment if it’s going to make people’s lives harder or add to workload, then you’re probably reducing the chances of that change happening.*
*(CCG-9, multiple CCGs)*

Thus, there was an agreement that AMS strategies need to fit within these constraints, for example, by being easily accessible and adaptable, not requiring (much) additional work or effort, and being concise and with clear benefits to practitioners. CCG professionals also reported the need for more tailoring of AMS approaches to practices and local contexts to account for the different reasons for high prescribing and variations in scope and areas for improvement. Practices were seen as diverse and independent organizations that should be supported in ways specific to their needs. CCG professionals thought that acknowledging and formalizing AMS as part of their roles and responsibilities, and more time and support (e.g., administrative) would allow them to work with practices more closely (e.g., visiting practices to provide training and collaboratively problem solve). Practice professionals also thought that additional financial resources and incentives would enable clinicians to prioritize and devote time specifically for AMS (e.g., auditing own and practice’s prescribing).


*I don’t think you can have a single approach. I think if you find that somebody isn’t following what you think is sensible advice, you need to go and have a conversation with them to understand what the barriers are to implement the advice that you think is sensible. Usually those barriers will have some substance. You then have to look at what the barriers are, you have to work out whether it’s possible to move those barriers.*
*(CCG-8, low-prescribing)*

### 3.3. Collaborative and Whole System Approaches

Participants stressed the importance of collaboration, effective communication, and leadership within organizations and across professional networks and settings. They recommended a whole system approach to AMS to promote greater consistency in antibiotic prescribing.

While the amount and ways of communicating varied largely within and between organizations, participants highlighted the importance of regular, effective communication. Interactive discussions, especially involving collaboratively identifying and addressing issues, were seen as challenging due to time constraints but more effective than passive communication (e.g., email newsletters). Participants described the value of discussing antibiotic prescribing within practice teams and between practices (e.g., in locality meetings), as well as within and between CCGs, which enabled sharing and learning from each other (e.g., about helpful strategies) and more consistent approaches.


*Sharing experience, sharing best practices and helping clinicians who are struggling is the only way. We already have a severe workforce crisis, we can’t have a GP leaving because they can’t deal with that.*
*(GP-15, low-prescribing)*


*I’d love to learn from those that have done really well and how they’ve achieved it because there is such a difference particularly with the London practices, I don’t know how they’re doing it. (…) It would be really interesting just to know how people have done it.*
*(CCG-4, high-prescribing)*

Participants discussed the importance of being connected and working with local networks. For CCG professionals, networks were often multidisciplinary and included GPs, hospital staff, microbiologists, pharmacists, out-of-hours providers, and other CCG teams (e.g., quality teams); for practice professionals, networks were with other local practices, microbiologists, and pharmacists. Poor communication with peers and networks and fragmentation of healthcare organizations were seen as barriers to AMS, whereas more teamwork, connectedness, and collaboration across networks and settings were seen as factors facilitating AMS. Participants stressed the importance of a whole system, consistent approach to AMS and antibiotics within and between organizations and settings (“singing from the same hymn sheet” (CCG-7, also GP-11 and GP-17)).


*I think it’s really important personally to have a multi-pronged approach so that you are all doing the same thing, and you don’t have a GP saying ‘No, you can’t have an antibiotic’ in their surgery, and then they walk to the local casualty at 7 pm and come home with Augmentin... It’s trying to have consistent messages from all areas of the NHS which is the hardest ask. *
*(CCG-8, low-prescribing)*

CCG participants reported the crucial role of national NHS leaders in promoting AMS as a national priority. Few leaders were particularly valued for raising the priority of AMS, influencing targets and agendas, organizing AMS networks, and acting as sources of helpful information and advice. Similarly, local AMS leaders and respected experts were seen as helpful in influencing professionals by sharing information and advice. They were particularly influential when seen as relevant specialists (e.g., microbiologists) and peer experts (e.g., GPs); the advice from peers was especially valued as it was perceived as more feasible in practice, rather than theoretical. Building connections and reputation within local networks were seen as important and helpful but took time to develop. 


*I have to say, the engagement from the Chief Medical Officer (Dame Sally Davies), I think that’s been very profound and supported what you’re trying to do on the ground. When you have such high-level leadership, it makes it a lot easier because there’s so many different things on the agenda of GPs, it’s sometimes hard to fight for space but when you’ve got buy-in at that level and also the Quality Premium, it makes shifting your thing up the agenda easier.*
*(CCG-4, high-prescribing)*


*We did a workshop where we got a clinician from another area. It was a doctor who had published about the work that he’d done in reducing antibiotic prescribing in his surgery and the approach that they’d taken and we got him to come and talk to all of our practices at a joint locality meeting because we felt that me standing there going, “you really ought to say no to antibiotics a bit more” probably wasn’t very helpful but a clinician who had done it, one of their peers was probably more helpful and that went down really well.*
*(CCG-2, medium-prescribing)*

Practice champions were also perceived as critical to quality improvement and practices taking “ownership of their own prescribing” (CCG-5). It was seen as important for the change to be driven from within the practices through a whole team approach working towards a common goal and led by practice leaders and champions. The role of CCG professionals was seen as supporting this process.


*…if a practice can be united in their approach to managing antimicrobial prescribing, then that seems to reflect a kind of lower overall antibiotic prescribing rate and that’s true of a lot of prescribing indicators. If you’ve got people working as a practice and not as a set of individuals, that’s really really helpful.*
*(CCG-9, multiple CCGs)*


*It’s something we’ve always been passionate about. Historically we’ve had a few GPs in particular, two or three of us who strongly believe in antibiotic stewardship. (…) I think the most important strategy is creating milieu in the practice to reduce unnecessary prescribing. If you ask me what I believe personally, you need to be well led. You need to have an environment in the practice that you’re all buying into that as a principle. And I think if you do that, you’ll be more successful than if you try to do it unilaterally, on your own sort of thing. You need to try and get buy-in. *
*(GP-6, low-prescribing)*

### 3.4. Relativity of Appropriate Prescribing 

Finally, participants’ views showed that appropriate antibiotic prescribing is, at least partly, relative and socially constructed as it depends on how it is assessed, what it is compared against, and on norms and expectations related to antibiotics.

Participants questioned how certain antibiotic prescribing targets and thresholds were identified and what evidence guided this process, describing them as “arbitrary”. Yet, they were then used to benchmark CCGs’ and practices’ prescribing, leading to defining some as “high” prescribers. Prescribing rates were seen as depending on how data was aggregated and averaged. For example, the apparent high prescribing could be due to including data from higher prescribing organizations (e.g., out-of-hours) or areas (e.g., with “different demographics” and “historically” different prescribing rates (CCG-9, multiple CCGs)). It could also be influenced by the prevalence of certain types of patients (e.g., older, children, with comorbidities) and clinics that may require more antibiotics, and therefore seen as justified rather than inappropriate.


*‘Why have they chosen 10%’? And why this and why that and, ‘They just plucked these figures out of the air’, so it did create some negative feedback. *
*(CCG-6, high-prescribing)*


*We’ve put in some arbitrary red, amber, green ratings on the dashboard for our antibiotics, and if people are green then they’re doing well, they’re performing to target, they’re doing what we want them to do, they wouldn’t have a great deal of additional scrutiny. If somebody is red and not moving at all, then that would be more cause for concern. *
*(CCG-8, low-prescribing)*

Perceptions of performance were also relative depending on whether prescribing rates were compared locally, nationally, or internationally. Being considered to be a high-prescribing CCG was often seen as relative to other CCGs that had also been improving but without taking into account different historical prescribing rates and contextual factors. Participants’ reports suggested that comparisons could motivate some organizations to optimize prescribing. However, they could also result in complacency (perceiving no need for further improvement) if favorable comparisons were made, or in disengagement (perceiving improvement as impossible) in cases of large disparity from targets or others’ performance. There was also a sense that some prescribers may try to circumvent the targets without optimizing antibiotics if they perceived no value in them. 


*For us it would be helpful if everybody else would stand still and give us a chance because we are reducing our antibiotics but equally so is everybody else so whilst we are lower than we were, we are still higher than other areas…. *
*(CCG-2, medium-prescribing)*


*Being the lowest percentage in the UK is still high when you compare it to other countries in the world.*
*(GP-6, low-prescribing)*


*This particular colleague who prescribes a lot of antibiotics used to prescribe a lot of Co-amoxiclav. Because the CCG are concentrating on that… he’s recently moved over to prescribing Azithromycin… He knows that this isn’t one that the CCG are picking up on and paying attention, so it’s a way of keeping his head down. He wants to prescribe an effective antibiotic which minimizes the risk of the patient getting worse and needing to be admitted to hospital and that’s his current way of doing it.*
*(GP-4, high-prescribing)*

Participants suggested that social norms around antibiotics and AMS among professionals and patients are changing. They reported that most healthcare professionals increasingly understand the importance of AMS and preserving antibiotics as part of good quality care. They described how most prescribers value feedback about their prescribing and consider adjusting it if it differs from others or guidelines without good reasons. High prescribers were referred to as “outliers”, “pockets of resistance” (CCG-1), or “rogues who are doing this inappropriate prescribing” (CCG-3), suggesting that high prescribing might be increasingly seen as socially unacceptable among professionals and as an issue of a minority of prescribers and organizations who might be blamed for their higher prescribing. CCG and practice professionals also described how patients’ awareness of the importance of not taking antibiotics unnecessarily had increased thanks to public health campaigns and prescribers’ education, suggesting that wider social norms about antibiotics among patients might be changing. 


*Generally, now, as prescribers and the general population seem to understand the responsibility that they have so generally I feel as though I’m pushing against an open door in supporting certainly the clinicians in trying to achieve appropriate prescribing in their practices. I think maybe that wouldn’t have been the case 10 or 15 years ago, but I think the perception has changed both in the public and other healthcare professionals. *
*(CCG-4, high-prescribing)*

## 4. Discussion

Our findings show how antibiotic prescribing and engagement with AMS in general practice in England is influenced by the social and contextual factors on multiple levels (individual, practice, local, and national) (Figure 1). Clinicians consider clinical and also social factors when making prescribing decisions, with variation in socioeconomic factors and health inequalities influencing the (actual and perceived) need for antibiotics. Moreover, practices that are struggling (e.g., with staff shortages, demand, and transitions) are seen as unable to engage with AMS, and all practices and CCGs are described as under pressure with many competing priorities and inadequate resources. In this context, AMS is seen as an additional burden and lower priority than immediate pressures, and prescribing antibiotics more freely is a way of dealing with the pressure. Underlying this is the perception of the arbitrariness of prescribing targets and data, and relativity of “appropriate” antibiotic prescribing and comparisons of prescribing rates. While targets and data are important change facilitators, in some situations they may be met with complacency, despondency, or disregard. Changing social norms and expectations about antibiotics mean that healthcare professionals and patients are increasingly aware of and perceive the importance of not using antibiotics unnecessarily. Participants believed that AMS and prudent prescribing may be encouraged by the following: tailoring of AMS initiatives to local contexts, ensuring acceptability and feasibility of AMS strategies that fit within situational constraints, leadership and teamwork, time for AMS, and a whole system approach to promote consistency and address wider health inequalities.

### 4.1. Comparison with the Literature

Socioeconomic deprivation has been linked with a higher prevalence of antibiotics in primary care across England [21], Scotland [22], Wales [23], and Northern Ireland [24], even after controlling for patient demographics, smoking, and chronic conditions. We found that deprived and ethnic minority populations were perceived as needing more antibiotics, as clinicians consider the wider context of patients’ lives, such as their (perceived) ability to self-care, reconsult if needed, risk of complications, and the impact of not working. This illustrates the role of perceptions of general practice clinicians in considering and treating patients holistically [25]. Participants also suggested that higher prescribing might be more likely in situations when explaining non-prescribing decisions to patients is difficult, for example, due to cultural understandings and spoken languages. While perceptions of patient expectations may be misinterpreted [26,27], research on how expectations for antibiotics may vary among different populations in England is lacking.

Substantial variability in antibiotic prescribing between practices remains unexplained by individual-level variables, with a wide variation in prescribing to less unwell patients, suggesting practice and prescriber factors [9,10,28]. Higher rates of antibiotics have also been correlated with more prescribing of other medicines [29]. Therefore, as we found, higher antibiotic prescribing might be indicative of some practice characteristics and contextual factors. Previous studies have found higher antibiotic prescribing to be associated with larger practices [7], non-training practices, shorter appointments, and GPs who were male, over 45 years old, and qualified outside the UK [12]. A qualitative study also found consultation length, triage system, and access to microbiologists or pharmacists to be perceived by GPs as influencing prescribing [17]. Additionally, we identified the influence of staff turnover and temporality, workload, competing priorities, and insufficient resources. In the pressured context, antibiotic (over)prescribing may be seen as a rational response to the circumstances (as found in hospitals [30]). With the perceived constant pressure on prescribers to change but no conducive change in the context, we found that there was a sense that further considerable change might be unrealistic. On the contrary, we found that practice culture and a wider context of good communication, teamwork, and leadership may facilitate better prescribing and engagement with AMS. 

Our findings highlight the role of AMS leaders and networks in sharing information, learning from each other’s experience, and raising the priority of AMS. Collaboration across professions, organizations, and settings, as well as a whole system approach was seen as needed to promote greater consistency in antibiotic prescribing. These findings are reflected in a survey of all CCGs which also found the perceived importance of relevant experts and champions and a whole practice, system wide approach [31,32], and challenges caused by insufficient time and other resources dedicated to AMS [33]. Leadership and championing, involvement in networks, and developing trusting, respectful relationships takes time. Ensuring adequate time and other resources is important for CCG professionals (acting as intermediaries supporting GPs [34]), and for practice professionals who may experience tensions between their clinical and leadership roles [35].

The influence of antibiotic-related social norms and perceived relativity of targets and comparisons were also apparent. Perceptions of others’ prescribing and support for no-antibiotic decisions seem to differentiate between low- and high-prescribing practices [17]. We also found that perceived lack of consistency may increase unnecessary prescribing to avoid patients re-consulting with another doctor or service (thus helping reduce workload). Moreover, we find that “appropriate” or “high” antibiotic prescribing is seen as relative as it depends on a comparator. Feedback on prescribing and benchmarking and comparing with other practices can influence social norms and help optimize antibiotics [18,32,36]. However, we found that in some cases it could also lead to complacency, despite a potential for optimizing antibiotics if compared to other reference points. It may also lead to disregard or disengagement, especially if targets are perceived as unachievable [18] or differences from others as too big [37]. Our participants highlighted that the sole focus on benchmarking does not take into account the different local contexts and changes that have been achieved within these. However, they reported that the wider social norms among professionals and patients related to appropriate antibiotic use have started to shift. Over time, this might help change the UK’s culture of high antibiotic prescribing to a “norm” to withhold antibiotics rather than prescribe [38].

Finally, despite many challenges, improvement in antibiotic prescribing (including high prescribers) needs acknowledging. Since the introduction of the QP antibiotic prescribing measures, most CCGs have met the targets [4]; moreover, the highest prescribing practices have reduced their antibiotic prescribing at a higher rate [39], and the variation across practices has been declining [7]. Therefore, now the focus needs to be on further supporting prescribers, practices, and commissioners in these improvements, and on reducing health inequalities by tailoring the support to the specific contexts and needs.

### 4.2. Implications

Figure 1 includes examples of possible implications for how the identified influences might be addressed to support antibiotic optimization. More research is needed to identify (novel) ways to address the interpersonal, organizational, and system-level influences on antibiotic use and health inequalities. More focus is also needed on how to adapt and implement existing interventions in ways that take into account the local context and fit within constraints (e.g., concise, with clear benefits). Adequate support and resources for CCGs and general practices would enable professionals to take time to focus on AMS. CCG professionals require more staff time and guidance to engage with practices more and provide support tailored to practice context and needs (e.g., to identify and address particular reasons for high prescribing). They may also be able to promote and reinforce antibiotic optimization among lower prescribers to sustain and make further improvements. Providing resources, such as protected time and additional funding, could mitigate some barriers and support AMS leaders and champions in CCGs and practices. A whole system approach (e.g., by involving leaders and networks across organizations and disciplines) is needed to promote consistency in prudent antibiotic prescribing and to address socioeconomic and health inequalities. 

### 4.3. Limitations

Purposeful sampling to ensure variation was planned but not possible; eventually, the final pragmatic sample was still relatively diverse and included CCGs and practices with a range of antibiotic prescribing levels, deprivation, and size. We included in the study all participants who expressed interest and agreed to participate; their views might have differed from those who did not volunteer to participate. As with all qualitative studies, the findings are time and context specific. 

## 5. Conclusions

Although antibiotic prescribing is an individual behavior, influenced by individual-level factors, social and contextual factors also impact prescribing. Deprivation, staff shortages and turnover, insufficient resources, and competing priorities were perceived as the main contextual barriers to prudent prescribing and engagement with AMS. Tailoring to local contexts, adequate resources, collaboration and teamwork, multidisciplinary networks, strong AMS leadership, and social norms among professionals and patients conducive to appropriate antibiotic use were seen as helping AMS promotion and engagement. Antibiotics will likely continue to be prescribed unnecessarily or inappropriately without changes in the context of prescribing, and without adequate time and resources to support accurate diagnosis and appropriate prescribing. Further optimization of antibiotic prescribing in primary care requires interventions and solutions that go beyond targeting individual behavior change and address social and contextual influences.

## Figures and Tables

**Figure 1 antibiotics-09-00859-f001:**
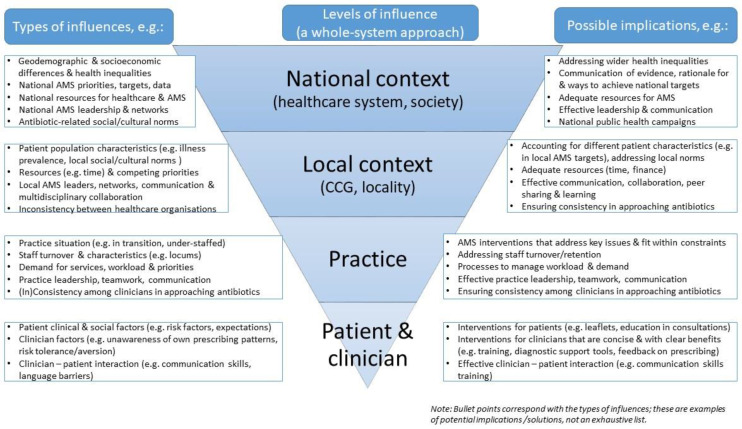
Summary of social and contextual influences on antibiotic prescribing and antimicrobial stewardship (AMS).

**Table 1 antibiotics-09-00859-t001:** Sample characteristics.

Characteristics	CCG Participants (*n* = 22, 33 CCGs)	General Practice Participants (*n* = 19)
Sex	17 females, 5 males	9 females, 10 males
Age (years)	35–60 (mean 48)	36–68 (mean 49)
Role	Leadership role, 11 Team member/prescribing advisor-type role, 11	General practitioners, 14 Nurse prescribers, 3 Practice managers, 2
Years in current organization	1–20 (mean 6)	1–35 (mean 13) General practitioners, 5–35, Nurse prescribers, 1–5, Practice managers, 7 and 10
Years in current role/since qualified	1–19 (mean 4)	2–45 (mean 21) General practitioners, 8–45 Nurse prescribers, 2 and 11
Size of CCG/general practice	9–97 practices (mean 40) Small (<25 practices), 10 CCGs Medium (25–75 practices), 20 CCGs Large (>76 practices), 3 practices	2–24 prescribers (mean 9) Small (2–5 prescribers), 7 practices Medium (6–15 prescribers), 11 practices Large (>20 prescribers), 1 practice
Deprivation ^1^	High (1–3 decile), 6 CCGs Medium (4–7 decile), 12 CCGs Low (8–10 decile), 15 CCGs	High (1–3 decile), 8 practices Medium (4–7 decile), 7 practices Low (8–10 decile): 4 practices
Antibiotic prescribing rates ^2^ (items/STAR-PU)	High (quintiles 4–5), 13 CCGs Medium (quintile 3), 9 CCGs Low (quintiles 1–2), 11 CCGs	High (>0.27), 4 practices Medium (0.25–0.27), 7 practices Low (<0.25), 8 practices

Notes: ^1^ Deprivation was based on the Index of Multiple Deprivation decile in England (2015); “high” deprivation level was considered for deciles 1–3, “medium” for deciles 4–7, “low” for deciles 8–10. ^2^ For CCGs, based on the PrescQIPP antibiotic prescribing data (items per STAR-PU for year 2017), the CCGs in the 1st or 2nd quintile of antibiotic prescribing in England were considered to be “low”, the CCGs in the 3rd quintile were considered to be “medium”, and the CCGs in the 4th or 5th quintile were considered to be “high”. For general practices, based on Fingertips data (items/STAR-PU for quarter 4, 2017), general practices with antibiotic prescribing rates under 0.25 were considered to be “low”, between 0.25 and 0.27 were considered to be “medium”, and over 0.27 were considered to be “high’. STAR-PU (specific therapeutic group age-sex related prescribing unit) is weighting used to take into account variation in the size and nature of the patient population.

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
