# Peer review of "Social and Contextual Influences on Antibiotic Prescribing and Antimicrobial Stewardship: A Qualitative Study with Clinical Commissioning Group and General Practice Professionals"

_antibiotics, 2020, doi:10.3390/antibiotics9120859_

Round 1
Reviewer 1 Report
The study aimed to assess social and contextual factors affecting antibiotic prescribing and engagement with antimicrobial stewardship (AMS) initiatives among CCG and general practitioners. The authors found a number of factors led to differences in prescribing patterns and AMS initiative adherence, depending on patient population, prescriber type, and region.
Overall, the manuscript is well written. There are some aspects that might be improved upon:
1) The example quotes that were included in the manuscript may be better suited for the supplemental material with the other quotes. In my opinion it somewhat disrupts the flow and makes the manuscript feel less of a research manuscript and more like a case series. I think along these lines, it may also be more beneficial to some how quantify the quotes in an objective manner that is less descriptive. Perhaps in a way that there is a numerical value associated with the specific observation being described, this could be done in text and/or table. Otherwise, it somewhat can seem that the authors pulled 1-2 examples to focus on, but it might bare more weight if there was a quantitative value assigned to each of those.
1a) Line 119, it would be helpful in defining or explaining how a theme was determined to be a "key theme" objectively from the study standpoint.
2) For readers outside of the United Kingdom or England, defining or explaining what CCGs are might be helpful to understand the differences between them and general practitioners. As someone outside England I had to research on CCGs, so maybe a short 1-2 sentences on what they are maybe helpful.
3) Table 1: certain rows, particularly "Size of ..." seems very busy with text in the boxes, can this be "cleaned up"?
4) Will the results to this study be followed up? It would seem as though based off of the "Key themes" identified a follow-up survey study using Likert scales for theme agreement would be possible. If so, this could be included in the discussion/conclusion section as future direction and plan.
Author Response
REVIEWER 1 comments
The study aimed to assess social and contextual factors affecting antibiotic prescribing and engagement with antimicrobial stewardship (AMS) initiatives among CCG and general practitioners. The authors found a number of factors led to differences in prescribing patterns and AMS initiative adherence, depending on patient population, prescriber type, and region.
Overall, the manuscript is well written. There are some aspects that might be improved upon.
- The example quotes that were included in the manuscript may be better suited for the supplemental material with the other quotes. In my opinion it somewhat disrupts the flow and makes the manuscript feel less of a research manuscript and more like a case series. I think along these lines, it may also be more beneficial to some how quantify the quotes in an objective manner that is less descriptive. Perhaps in a way that there is a numerical value associated with the specific observation being described, this could be done in text and/or table. Otherwise, it somewhat can seem that the authors pulled 1-2 examples to focus on, but it might bare more weight if there was a quantitative value assigned to each of those.
We appreciate that this style of reporting might seem unusual to the Reviewer or those less familiar with qualitative research. However, the reporting of themes and quotes, and not quantifying the quotes, is typical of qualitative studies. In qualitative studies the quotes are the data that support the findings and authors’ interpretations, and therefore are a critical part of good quality qualitative papers. They are required in good quality qualitative papers (as recommended in reporting standards for qualitative papers, e.g. item S17 ‘links to empirical data’ in the O’Brien et al. 2014 ‘Standards for Reporting Qualitative Research: A synthesis of recommendations’).
Quantifying quotes is not an approach taken in qualitative research and a lack of numerical values is not an indication of ‘subjectivity’ or poor quality of qualitative studies. The importance of the themes is not determined on the basis of numbers but on their relevance to addressing qualitative research questions of ‘how’ and ‘why’. Quantitative study designs are better suited to research questions about prevalence. In our study, we followed a systematic approach to qualitative data analysis (inductive thematic analysis, as outlined by Braun and Clarke, 2006). Our methods and reporting were in accordance with the methods and reporting standards of thematic analysis.
- Line 119, it would be helpful in defining or explaining how a theme was determined to be a "key theme" objectively from the study standpoint.
Key themes were those that related to the research question and were supported by the analysed data (i.e. coded and categorised), and then agreed on through discussions with the study team members. We have clarified this in the paper (lines 124-127).
- For readers outside of the United Kingdom or England, defining or explaining what CCGs are might be helpful to understand the differences between them and general practitioners. As someone outside England I had to research on CCGs, so maybe a short 1-2 sentences on what they are maybe helpful.
We have included a sentence explaining what CCGs are (lines 56-58).
- Table 1: certain rows, particularly "Size of ..." seems very busy with text in the boxes, can this be "cleaned up"?
We have simplified the reporting of the size of CCGs/practices. We are unsure to reduce other details without losing any data reported.
- Will the results to this study be followed up? It would seem as though based off of the "Key themes" identified a follow-up survey study using Likert scales for theme agreement would be possible. If so, this could be included in the discussion/conclusion section as future direction and plan.
Currently we have no plan to follow up this qualitative study with a quantitative study. As reported in the Introduction and Discussion (‘Comparison with the literature’) some of the reported influences have also been already identified as associated with variability in antibiotic prescribing by quantitative, cross-sectional studies (e.g. deprivation, practice size, appointment length, prescriber characteristics). Our study attempted to add to these studies by showing how (i.e. the mechanisms/processes how) these contextual and social factors are perceived to influence antibiotic prescribing.
Reviewer 2 Report
The authors attempt to tackle a complex, yet important issues of factors that influence the prescription of antibiotics. Their focus on social and contextual factors is important, and the results of any such investigations having the potential to alter prescribing practices. Yet, and sad to say, little new information has been presented that would remedy the situation. Nonetheless, publication of information such as being presented in this paper, is important to emphasize the need for more accurate prescription of antibiotics.
The following suggestions are made to improve the study and the paper:
1-While quotations from practitioners are valuable, the numbers of those need to be reduced and be less repetitive.
2-The overall length of the paper should be reduced, as its current length will be distracting to readers.
3-There is a need for better focus on the actual differences, if any, of more experienced providers and those with more advanced degrees v those without medical degrees. In this regard, it seems confounding that "older" GPs would prescribe more antibiotics than others; this needing better detailing and understanding. Also needed is an analysis of the effect of post-graduate courses on prescribing practices, with the obvious needed statement in the conclusion that such courses should be available and strongly recommended for all practices.
4-There is no indication in the research protocol of the possible effect of practitioners prescribing more antibiotics for fear of malpractice allegations.
5-The reality check here is that without the time and tools to ensure an accurate diagnosis upon which to prescribe an antibiotic, there will continue to be practice to prescribe antibiotics when they might not be necessary.
Author Response
REVIEWER 2 comments
The authors attempt to tackle a complex, yet important issues of factors that influence the prescription of antibiotics. Their focus on social and contextual factors is important, and the results of any such investigations having the potential to alter prescribing practices. Yet, and sad to say, little new information has been presented that would remedy the situation. Nonetheless, publication of information such as being presented in this paper, is important to emphasize the need for more accurate prescription of antibiotics.
Thank you to the Reviewer for recognising the importance of focusing on identifying and understanding social and contextual influences on antibiotic prescribing. Much research so far has reported on more immediate influences on individual prescribers, whereas our focus has been broader and considered the wider context in which prescribers work. We propose implications that would potentially help address these influences throughout the results (i.e. those reported by the participants) and in the right hand-side column in Figure 1 (where each bullet point suggests a possible implication that relates to each identified social/contextual influence). However, we take the point that the study does not offer ‘information… that would remedy the situation’ – unfortunately these influences are very complex and complicated, and our study did not aim at identifying novel solutions to these influences but rather at understanding how they influence antibiotic prescribing. We have amended this text at the start of the Implication section to clarify this point and to highlight the proposed implications in Figure 1 (lines 519-523).
The following suggestions are made to improve the study and the paper.
- While quotations from practitioners are valuable, the numbers of those need to be reduced and be less repetitive.
- The overall length of the paper should be reduced, as its current length will be distracting to readers.
We appreciate that the paper is a bit longer than papers typically targeted at healthcare professionals but, at the same time, qualitative papers are typically longer due to the inclusion of quotes. The quotes are data supporting the findings and authors’ interpretations (thus it may seem that they repeat the text), and therefore are a critical part of good quality qualitative papers (and recommended in the standards for reporting of qualitative studies). We were selective and included only the most appropriate and illustrative quotes, keeping many other supportive quotes in supplementary materials.
- There is a need for better focus on the actual differences, if any, of more experienced providers and those with more advanced degrees v those without medical degrees. In this regard, it seems confounding that "older" GPs would prescribe more antibiotics than others; this needing better detailing and understanding. Also needed is an analysis of the effect of post-graduate courses on prescribing practices, with the obvious needed statement in the conclusion that such courses should be available and strongly recommended for all practices.
We do not mean to suggest that every ‘older’ GP would prescribe more antibiotics than average. Our findings describe the perceptions of interviewees that some older GPs seem more likely to prescribe more antibiotics due to long-standing habits of prescribing antibiotics (e.g. more readily than not prescribing, or prescribing specific types of antibiotics perhaps no longer recommended as first-line treatment). This might relate to the time since their medical training/qualification rather than age per se. Wang et al. (2009, BJGP) have found in their cross-sectional analysis that in UK practices (among other factors) higher proportion of GPs over 45 years old was associated with practices prescribing more antibiotics (reference 12). We have clarified this section of the results (lines 213-217).
We agree that training courses might be used as a strategy to promote/ensure appropriate antibiotic prescribing. We describe such strategies in another paper where we focused on how the Quality Premium was implemented (reference 18). Since this is not so much a contextual/social influence, and is instead concerned with interventions targeting individual knowledge and skills, we did not report and expand on this aspect in this paper.
- There is no indication in the research protocol of the possible effect of practitioners prescribing more antibiotics for fear of malpractice allegations.
We agree with the Reviewer that there is a possibility of prescribers prescribing more antibiotics when they may perceive the risk of malpractice allegations and legal consequences when not prescribing, and we have found this in our data. We have clarified this on line 232.
- The reality check here is that without the time and tools to ensure an accurate diagnosis upon which to prescribe an antibiotic, there will continue to be practice to prescribe antibiotics when they might not be necessary.
Thank you for your comment that captures the implications of our findings. We have added another sentence in the conclusions to further stress this point (lines 548-550).
Reviewer 3 Report
Antibiotic prescribing is an interesting topic for the medical community. The way general practicioners apply it is also of interest.
This particular study is very descriptive and I think it should be published in another kind of journal
Author Response
REVIEWER 3 comments
Antibiotic prescribing is an interesting topic for the medical community. The way general practicioners apply it is also of interest.
This particular study is very descriptive and I think it should be published in another kind of journal.
It appears that the Reviewer refers to our study as ‘descriptive’ because it is a qualitative, interview study. Qualitative methodologies are recognised approaches to study antibiotic prescribing and are particularly suitable to address research questions that help better understand how and why different factors might influence clinician and patient behaviours. Therefore, we believe our study makes a helpful contribution to the research, and extends the quantitative studies on factors associated with antibiotic prescribing (as we explained in the Introduction and Discussion). Qualitative research ‘exploring the determinants of antimicrobial use and resistance’ is listed as within the scope of Antibiotics and qualitative studies, such as ours, have been published in Antibiotics.
Round 2
Reviewer 2 Report
The authors have responded appropriately to the comments. The readability could still be improved by limiting the number of individual responses of the interviewees.
Reviewer 3 Report
The changes made by the authors are satisfactory, I have no other comments